# Immune Cell Engagers: Advancing Precision Immunotherapy for Cancer Treatment

**DOI:** 10.3390/antib14010016

**Published:** 2025-02-11

**Authors:** Hyukmin In, Minkyoung Park, Hyeonsik Lee, Kyung Ho Han

**Affiliations:** Department of Biological Sciences and Biotechnology, Hannam University, Daejeon 34054, Republic of Korea

**Keywords:** immune cell engagers, T-cell engagers, NK cell engagers, phagocyte cell engagers

## Abstract

Immune cell engagers (ICEs) are an emerging class of immunotherapies designed to harness the immune system’s anti-tumor potential through precise targeting and activation of immune effector cells. By engaging T cells, natural killer (NK) cells, and phagocytes, ICEs overcome challenges such as immune evasion and MHC downregulation, addressing critical barriers in cancer treatment. T-cell engagers (TCEs), led by bispecific T-cell engagers (BiTEs), dominate the field, with innovations such as half-life-extended BiTEs, trispecific antibodies, and checkpoint inhibitory T-cell engagers driving their application in hematologic and solid malignancies. NK cell engagers (NKCEs) and phagocyte cell engagers (PCEs) are rapidly progressing, drawing on NK cells’ innate cytotoxicity and macrophages’ phagocytic abilities to target tumors, particularly in immunosuppressive microenvironments. Since the FDA approval of Blinatumomab in 2014, ICEs have transformed the oncology landscape, with nine FDA-approved products and numerous candidates in clinical trials. Despite challenges such as toxicity, resistance, and limited efficacy in solid tumors, ongoing research into advanced platforms and combination therapies highlights the growing potential of ICEs to provide personalized, scalable, and effective cancer treatments. This review investigates the mechanisms, platforms, research trends, and clinical progress of ICEs, emphasizing their pivotal role in advancing precision immunotherapy and their promise as a cornerstone of next-generation cancer therapies.

## 1. Introduction

In the 20th century, advancements in immunology and molecular biology established immunotherapy as a revolutionary field in medicine. Discoveries such as cytokines, including interferons and interleukins, highlighted the importance of key molecules in regulating immune responses, while monoclonal antibodies enabled precise targeting of specific antigens. Furthermore, the identification of immune checkpoint proteins like CTLA-4 and PD-1 unveiled critical pathways by which cancer cells evade immune surveillance, and the development of immune checkpoint inhibitors revolutionized cancer treatment paradigms. Immune Cell Engagers (ICEs) are molecule-based therapies designed to harness the immune system’s anti-tumor mechanisms by activating immune cells to target cancer cells, thereby overcoming immune evasion and redirecting immune cells for enhanced anti-tumor activity [1,2]. Unlike conventional chemotherapy and radiotherapy, which can harm normal tissues, ICEs induce tumor-specific immune responses with minimized off-target effects, making them a promising therapeutic modality [3,4].

ICEs are classified based on the type of immune effector cells they engage: T-cell engagers (TCEs), natural killer cell engagers (NKCEs), and macrophage engagers. T-cells, among the most potent anti-cancer immune cells, are central to many ICE strategies. Bispecific T-cell engagers (BiTEs) are a prominent example, where one single-chain variable fragment (scFv) binds tumor-associated antigens (TAAs) on cancer cells, while the other targets CD3 on T-cell receptors (TCR) [5]. BiTEs facilitate the formation of immune synapses, enabling T-cells to release perforin and granzymes to kill tumor cells. Compared to chimeric antigen receptor T-cell (CAR-T) therapies, BiTEs have advantages such as simplified production and scalability for mass production without requiring patient-specific customization Trispecific antibodies (TsAbs) further enhance T-cell activation by combining T-cell engagement with co-stimulatory signals or immune checkpoint inhibition [6].

NKCEs utlize NK cells, innate immune effectors that eliminate tumor cells independently of MHC recognition. NKCEs bind NK cells to tumor cells via activation receptors such as CD16a, inducing potent anti-cancer responses [7,8]. Bispecific NK cell engagers (BiKEs) and trispecific antibodies (TsAbs) are being investigated to enhance NK cell activity, with TsAbs incorporating cytokines like IL-15 to boost efficacy [9,10]. CAR-NK therapies represent another innovative approach, genetically engineering NK cells to express chimeric antigen receptors (CARs) for tumor-specific targeting. CAR-NKs offer advantages over CAR-Ts, including reduced risks of cytokine release syndrome (CRS), neurotoxicity, and off-target effects due to their shorter lifespan [11,12].

Macrophages, key players in the tumor microenvironment (TME), also play a critical role in ICEs through CAR-M (CAR-macrophage) therapies. CAR-Ms express CARs that enable macrophages to phagocytose tumor cells and present tumor antigens, activating adaptive immunity. They can also reprogram tumor-promoting M2 macrophages into pro-inflammatory M1 macrophages, enhancing tumor-specific T-cell responses [3,13]. A CAR-M therapy targeting HER2 is currently under evaluation in a phase 1 clinical trial [14].

ICEs are being developed across diverse platforms, with bispecific antibodies achieving rapid commercialization [4]. Emerging strategies, including nanobody-based ICEs and therapies that exploit efferocytosis to modulate inflammatory responses, are also under development [15]. Since the FDA approval of blinatumomab (Blincyto) for acute lymphocytic leukemia in 2014, subsequent ICEs targeting hematologic cancers, such as mosunetuzumab, epcoritamab, and teclistamab, have entered clinical use. Blincyto alone achieved approximately USD 861 million in sales in 2023, and the ICE market, encompassing bispecific antibodies, is expected to grow into billions of dollars by the late 2020s. To date, BiTEs have achieved the most clinical success, with nine products approved by the FDA and EU as of August 2023. In contrast, TsAbs, NKCEs, and macrophage-engaging agents remain in earlier stages of research, with many undergoing early-phase clinical trials [9].

This review aims to provide a comprehensive overview of ICEs, detailing their classifications, mechanisms of action, and recent research trends. By highlighting advancements in ICE development, this article seeks to support innovation in next-generation immunotherapy, facilitate the expansion of ICE indications, and contribute to the development of therapies with improved specificity and efficacy.

## 2. Mechanisms of Immune Cell Engagers (ICEs)

Tumor immunity involves a coordinated effort between innate and adaptive immune responses to detect and eliminate malignant cells [16]. ICEs, a rapidly advancing class of therapeutics, exploit these immune mechanisms by redirecting immune effector cells, such as T cells, natural killer cells, and phagocytic cells, toward tumor eradication, even within the immunosuppressive tumor microenvironment (TME) [9]. Understanding the detailed mechanisms of action of these immune cell subsets and how ICEs optimize their activity is pivotal for advancing cancer immunotherapy (Figure 1).

T cells are the central players in adaptive immunity, particularly cytotoxic T lymphocytes, which recognize tumor-associated antigens presented on MHC molecules by tumor cells. Upon engagement, T cells initiate a cascade of intracellular signaling that leads to their activation, proliferation, and cytotoxic functions. Activated CTLs release granules containing perforin, which forms pores in the target cell membrane, and granzymes, which enter through these pores to trigger caspase-mediated apoptosis. Additionally, CTLs utilize Fas–FasL interactions to induce apoptosis via the extrinsic pathway [17].

TCEs function by binding simultaneously to tumor-associated antigens (TAAs) on tumor cells and the CD3 complex on T cells, creating an immune synapse that activates T cells and promotes tumor cell destruction. Bispecific T-cell engagers (BiTEs), a key TCE platform, utilize single-chain variable fragments (scFvs) linked to target TAAs and CD3. This design efficiently bridges T cells and tumor cells, triggering immune synapse formation and cytotoxic activity without requiring co-stimulatory signals like CD28 or IL-2 [18,19,20]. BiTEs reawaken exhausted T cells and initiate robust antitumor responses by stimulating the release of cytotoxic molecules such as perforin and granzymes. With their rapid kinetics and non-MHC-dependent activation, TCEs enable bystander killing, allowing T cells to target both TAA-positive and adjacent TAA-negative tumor cells. This mechanism minimizes antigen escape and enhances therapeutic efficacy, making TCEs a cornerstone of modern tumor immunotherapy [5,9].

NK cells, innate lymphocytes with natural cytotoxic potential, are critical in recognizing and eliminating malignant cells without prior antigen sensitization. They identify stressed or transformed cells through activating receptors like NKG2D or CD16, which binds the Fc region of antibodies in antibody-dependent cellular cytotoxicity (ADCC). Once activated, NK cells release cytotoxic granules containing perforin and granzymes, while secreting cytokines such as IFN-γ to enhance antitumor immunity [17].

NK cell engagers exploit these mechanisms by cross-linking tumor cells with activating NK cell receptors. For example, NKCEs targeting CD16a enhance ADCC, augmenting NK cell degranulation and cytokine release. This activation is independent of MHC or TCR, enabling NKCEs to function effectively in tumors with low immunogenicity or MHC downregulation. Additionally, NKCEs stimulate the production of chemokines, attracting other immune cells to the tumor site [7,8].

Phagocytic cells, including macrophages and dendritic cells, contribute to tumor immunity through phagocytosis, antigen presentation, and cytokine secretion. Macrophages, particularly tumor-associated macrophages (TAMs), are polarized into pro-tumorigenic M2 or anti-tumorigenic M1 phenotypes. While M2 TAMs promote tumor progression and immune suppression, reprogramming them into M1-like macrophages can reinvigorate antitumor responses. M1 macrophages enhance phagocytosis, ROS production, and pro-inflammatory cytokine release (e.g., TNF-α, IL-12) [21,22].

Phagocytic cell engagers target macrophages to induce M1 polarization and enhance antibody-dependent cellular phagocytosis (ADCP). By engaging Fcγ receptors or specific TAM markers, PCEs facilitate the engulfment of opsonized tumor cells and promote antigen presentation to T cells. Additionally, dendritic cells activated by PCEs improve antigen processing and presentation, further amplifying T cell-mediated responses [14,23,24].

The TME significantly influences the efficacy of immune responses and ICEs. It is characterized by hypoxia, high lactate levels, and immune suppression mediated by regulatory T cells and M2 macrophages. Tumor cells exploit these conditions to evade immune detection and suppress effector cell functions. However, ICEs operate effectively within the TME by redirecting immune cells to target tumors while overcoming inhibitory signals. For instance, TCEs and NKCEs activate immune cells independent of MHC or co-stimulatory signals, while PCEs reprogram suppressive macrophages and enhance antigen presentation [25,26].

ICEs offer a highly specific approach to targeting tumors by binding TAAs and engaging immune effector cells. Their MHC-independent mechanisms overcome challenges associated with antigen presentation deficits in tumors. Furthermore, the ability of ICEs to induce bystander killing reduces the risk of tumor escape due to antigen heterogeneity. However, the widespread activation of immune cells can lead to adverse effects, such as cytokine release syndrome and off-target toxicity [4]. Additionally, the immunosuppressive TME remains a significant barrier that must be addressed in conjunction with ICE therapies.

Outlines the mechanism of action of ICEs based on bispecific antibodies, the most extensively studied format. The primary immune effector cells targeted by ICEs include T cells, NK cells, and phagocytic cells. ICEs simultaneously recognize immune cell surface receptors such as CD3, CD16a, and CD64, along with tumor-associated antigens (TAAs), to form an immunological synapse and activate immune cells. The formation of this synapse occurs independently of MHC recognition and reduces the physical distance between cells, facilitating ligand recognition by other immune cell receptors. Activated immune cells release cytokines such as IFN-γ and TNF-α, as well as cytotoxic molecules like perforin and granzyme, to direct immune effector cell responses and induce tumor cell cytotoxicity. During this process, the bystander effect may occur, where the death of tumor cells leads to the elimination of nearby tumor cells.

## 3. Antibody Platforms of Immune Cell Engagers

Advances in antibody engineering have unlocked new potential in cancer immunotherapy, with emerging platforms like bispecific antibodies, single-chain fragment variable complexes, and nanobodies leading the charge. These technologies leverage unique structural and functional designs to enhance immune targeting, stability, and therapeutic efficacy. BsAbs create precise immune synapses by binding tumor-associated antigens and immune cell receptors simultaneously, enabling targeted cancer cell elimination. Meanwhile, scFv complexes offer versatility with their compact structure and customizable linkers, and nanobodies stand out for their tissue penetration and access to hidden epitopes. Together, these innovations redefine the landscape of ICEs, addressing critical challenges in treating both solid and hematologic malignancies.

### 3.1. Bispecific Antibodies (BsAbs)

BsAbs are engineered to bind simultaneously to tumor-associated antigens on cancer cells and receptors on immune cells, such as CD3 on T cells or CD16a on NK cells. This dual targeting facilitates the formation of immune synapses, enabling precise immune activation and the targeted elimination of tumor cells (Figure 1). Modern BsAbs use IgG-based or variable fragment backbones to improve stability, functionality, and production efficiency. These designs allow for effective application across both solid and hematologic cancers [5].

### 3.2. Single-Chain Fragment Variable (scFv) Complex

scFv complexes are composed of variable heavy (VH) and light (VL) chain regions connected by a flexible peptide linker [27]. These complexes ensure proper folding and stability of antibodies. The composition of the linker influences solubility and functionality, with hydrophilic residues enhancing folding efficiency. Although scFvs are versatile and widely used in ICEs, their thermal instability can be mitigated through structural modifications like introducing disulfide bonds [28].

### 3.3. Nanobody

Nanobodies derived from camelid heavy-chain-only antibodies are the smallest functional antibody fragments with remarkable antigen-binding specificity and stability. Their compact size allows them to penetrate tissues effectively, making them ideal for targeting solid tumors. However, their rapid clearance and short serum half-life present challenges, which can be addressed by coupling nanobodies with other therapeutic agents or delivery systems. Nanobodies are uniquely capable of binding to hidden epitopes that conventional antibodies cannot access, adding significant value to cancer immunotherapy. These structural innovations enhance the capabilities of ICEs, providing customized solutions for complex therapeutic challenges in cancer treatment [29,30].

## 4. T Cell Engagers (TCEs)

TCEs are bispecific antibody-based therapies designed to address the challenge of tumor cells downregulating MHC expression, a common mechanism of immune evasion that disrupts T-cell-mediated antitumor activity [31,32]. These therapies function through MHC-independent mechanisms, directly targeting the T-cell receptor or CD3 complex on T cells and tumor-associated antigens (TAAs) on cancer cells. This binding activates T cells, inducing the release of perforin and granzymes, which eliminate tumor cells. Advanced TCE platforms include trispecific TCEs that incorporate additional features to enhance T-cell activation and half-life-extended TCEs that use Fc domains to prolong circulation time in vivo. These therapies are characterized by high tumor specificity, rapid immune responses, and the ability to target tumors even with low T-cell numbers. Their versatility and potential to integrate next-generation designs, such as triple-specific antibodies (TsAbs), have established TCEs as critical tools in modern cancer immunotherapy [32].

### 4.1. Bispecific T-Cell Engagers (BiTEs)

BiTEs have gained significant attention, particularly following the clinical success of blinatumomab in treating acute lymphoblastic leukemia (ALL). However, challenges such as inconvenient administration, resistance, and limited efficacy in solid tumors have spurred efforts to improve BiTE designs and develop multifunctional T-cell engaging antibodies. Modified BiTE constructs and novel TCE platforms are currently being explored in clinical trials, highlighting their potential to address these challenges (Figure 2) [19,33].

#### 4.1.1. Half-Life-Extended BiTEs (HLE-BiTEs)

HLE-BiTEs represent a significant advancement in cancer immunotherapy, addressing the pharmacokinetic limitations of traditional BiTE molecules. By incorporating an Fc domain that interacts with the neonatal Fc receptor (FcRn), HLE BiTEs achieve prolonged serum half-life, reducing dosing frequency and enhancing patient convenience (Figure 3A). These agents bind simultaneously to tumor-associated antigens, such as CD33 in acute myeloid leukemia (AML), and CD3 on T cells, facilitating robust T-cell-mediated cytotoxicity. Clinical trials have shown that HLE BiTEs, such as AMG 673 and AMG 701, maintain potent antitumor activity while reducing the need for continuous infusion therapy. However, challenges such as cytokine release syndrome and potential toxicity from prolonged serum concentrations necessitate optimized dosing strategies. The extended half-life and potential for combination therapies position HLE BiTEs as a promising approach in treating hematological malignancies and solid tumors [19,34,35,36,37].

#### 4.1.2. Checkpoint Inhibitory T-Cell Engagers (CiTEs)

CiTEs are a novel class of immunotherapeutic agents that combine bispecific T-cell engager technology with immune checkpoint blockade to enhance T-cell-mediated antitumor activity (Figure 3B). By targeting tumor-associated antigens, such as CD33 or EGFR, while simultaneously inhibiting checkpoint pathways like PD-1/PD-L1 or CTLA-4, CiTEs overcome immune suppression within the tumor microenvironment and restore T-cell activation. The integration of the extracellular domain of PD-1 enables CiTEs to block PD-L1 with reduced off-target effects, enhancing T-cell activation, increasing IFN-γ production, and inducing tumor cell lysis [38,39]. Preclinical studies demonstrate their superior antitumor efficacy compared to traditional BiTEs, with potential applications in hematologic malignancies and solid tumors. Despite promising results, challenges such as off-target effects, cytokine release syndrome, and the need for optimized checkpoint blockade remain areas of active investigation. Current research focuses on refining CiTE design and evaluating their clinical performance, positioning them as promising candidates for treating resistant or refractory cancers [40,41].

#### 4.1.3. Simultaneous Multiple Immune Targeting Engagers (SMiTEs)

SMITEs are an advanced immunotherapeutic strategy designed to overcome the limitations of single-target approaches by engaging multiple immune pathways (Figure 3C). These engineered molecules simultaneously bind tumor-associated antigens and immune receptors, such as CD3 on T cells, while incorporating co-stimulatory signals through molecules like 4-1BB (CD137), OX40 (CD134), or CD28. This multi-targeting approach enhances T-cell activation, proliferation, and survival, amplifying antitumor responses and overcoming immune suppression in the tumor microenvironment [42,43]. Preclinical models demonstrate that SMITEs induce more robust and sustained immune responses, reducing tumor burden more effectively than single-target therapies or conventional BiTEs. Despite their potential, challenges such as balancing immune activation to minimize toxicity, cytokine release syndrome, and complex molecular design remain areas for refinement. Ongoing research aims to optimize SMITE constructs and evaluate their applications in hematologic and solid malignancies, highlighting their transformative potential in cancer immunotherapy [44,45,46].

#### 4.1.4. Secreted Bispecific T-Cell Engagers

Secreted BiTEs designed to localize therapeutic activity within the tumor microenvironment, improving efficacy and reducing systemic toxicity (Figure 3D). Unlike systemically administered BiTEs, secreted BiTEs are expressed and released at the tumor site by genetically modified cells, such as CAR T cells, oncolytic viruses (OVs), or tumor-targeting bacteria. This localized secretion enhances BiTE concentration at the tumor, enabling precise activation of T cells via CD3 binding and simultaneous targeting of tumor-associated antigens (TAAs) like HER2 or EGFR [47,48]. Preclinical studies demonstrate that secreted BiTEs maintain therapeutic concentrations, minimize off-target effects, and overcome challenges such as rapid systemic clearance associated with traditional BiTEs. Combining secreted BiTEs with CAR T cells or OVs has shown synergistic efficacy, further reducing resistance in solid tumors. Despite challenges in optimizing delivery systems and secretion durability, ongoing research highlights their transformative potential in treating hematologic and solid malignancies [31,49,50].

#### 4.1.5. T-Cell Engagers with Silenced Fc Domains

T-Cell Engagers with Silenced Fc Domains engineered to enhance the safety and efficacy of T-cell-mediated cancer therapy by minimizing off-target effects (Figure 3E). Unlike traditional T-cell engagers, which include Fc domains capable of interacting with Fcγ receptors on immune cells, these modified molecules feature genetically or chemically silenced Fc domains to prevent non-specific immune activation, cytokine release syndrome, and rapid clearance. By specifically binding tumor-associated antigens (e.g., CD19 or EGFR) and CD3 on T cells, they activate cytotoxic responses precisely at the tumor site, reducing systemic toxicity. Preclinical studies and clinical trials, such as NCT03625037, have demonstrated that these modifications improve therapeutic outcomes by enhancing T-cell activation and infiltration at tumor sites while minimizing off-target effects. Ongoing research is focused on optimizing silenced Fc designs for stability, scalability, and efficacy in hematologic and solid malignancies, positioning them as a promising tool in cancer immunotherapy [32,51,52].

#### 4.1.6. Multivalent and Multispecific T-Cell Engagers

Multivalent and Multispecific TCEs are a next-generation approach in cancer immunotherapy, designed to enhance efficacy by simultaneously targeting multiple tumor-associated antigens (TAAs) and immune pathways (Figure 3F). Unlike traditional BiTEs that target a single TAA and CD3, multivalent TCEs integrate additional valencies and co-stimulatory receptors, such as 4-1BB (CD137) or CD28, to improve T-cell activation, proliferation, and cytotoxicity while addressing antigen heterogeneity and reducing immune escape. These constructs demonstrate higher binding affinity, expanded TAA recognition, and reduced off-target effects by achieving greater tumor specificity [53,54]. Preclinical studies, including triple-specific TCEs targeting CD3, CD28, and CD38, have shown superior tumor-killing potency and sustained immune responses in complex cancers such as multiple myeloma. Despite challenges such as molecular complexity, potential toxicity, and dosing strategies, ongoing clinical trials highlight the transformative potential of multivalent TCEs in treating hematologic and solid malignancies [55,56].

These advanced TCE platforms, including HLE-BiTEs, CiTEs, and SMiTEs, secreted BiTEs, silenced Fc domain TCEs, and multispecific constructs, represent the continuous evolution of T-cell immunotherapy. By addressing the limitations of conventional BiTEs, these innovations expand the therapeutic potential of TCEs across a wide range of malignancies.

## 5. NK Cell Engagers (NKCEs)

NKCEs are a novel class of immunotherapeutic agents designed to use the innate immune capabilities of natural killer (NK) cells for targeted cancer therapy. By bridging NK cells to tumor-associated antigens (TAAs) such as HER2, CD19, or BCMA, NKCEs enhance NK cell-mediated cytotoxicity through the activation of receptors like CD16 (FcγRIII), NKG2D, or NKp30 (Figure 4). This interaction facilitates immune synapse formation, leading to the release of cytotoxic granules and cytokines like IFN-γ, effectively lysing tumor cells. Unlike T and B cells, NK cells do not require antigen presentation or human leukocyte antigen (HLA) restrictions, making them less susceptible to tumor immune evasion [57,58,59]. They balance activating and inhibitory signals, such as stress-induced ligands or the absence of MHC-I, to distinguish abnormal cells from healthy ones. Activation occurs through cross-linking of receptors, triggering ITAM phosphorylation and downstream cytotoxic signaling cascades. Preclinical studies highlight NKCEs’ effectiveness against a broad range of malignancies, including those resistant to T-cell-based therapies. Challenges such as immune suppression within the tumor microenvironment and low NK cell infiltration remain, but ongoing research into combination therapies and delivery optimization aims to enhance their efficacy [60,61,62,63].

### 5.1. Types of NK Cell Engagers (NKCEs)

NKCEs are categorized based on structural complexity as bispecific killer cell engagers or trispecific killer cell engagers. Bispecific killer cell engagers are composed of two single-chain variable fragments connected by a linker. One fragment targets an activation receptor like CD16a on NK cells, while the other binds a tumor-associated antigen on cancer cells. This structure bridges NK cells and tumor cells, enhancing cytotoxicity and enabling precise tumor elimination [7,62]. Trispecific killer cell engagers build on bispecific designs by incorporating a third domain, such as a cytokine like IL-15, to enhance NK cell proliferation, survival, and cytotoxic activity. These constructs activate multiple receptors simultaneously, improving tumor targeting and immune responses. Trispecific engagers are particularly effective in hematologic malignancies where NK cells play a critical role [7,9]. NKCEs reduce the distance between NK cells and tumor cells, enabling efficient targeting. AFM13, an NKCE targeting CD30 on tumor cells and CD16a on NK cells, has demonstrated promising results in clinical trials for Hodgkin lymphoma and CD30-positive solid tumors [62,63].

### 5.2. Target Receptors for NK Cell Engagers (NKCEs)

NK cell activation is regulated by a balance of activating and inhibitory signals to ensure immune homeostasis and prevent unintended tissue damage. Key activating receptors on NK cells include C-type lectin receptors such as CD94/NKG2C and NKG2D, natural cytotoxicity receptors like NKp30 and NKp46, and killer cell C-type lectin-like receptors such as NKp65. Among these, CD16a, NKp46, NKp30, NKG2C, and NKG2D are the primary targets for NKCE-based immunotherapies [64] (Figure 5).

#### 5.2.1. CD16a

CD16a is a highly expressed receptor on NK cells and the most extensively studied target for antibody-dependent cellular cytotoxicity (ADCC). When CD16a binds to the Fc region of antibodies, it phosphorylates ITAMs on the CD3ζ and FcϵRIγ chains through Src family kinases, initiating downstream signaling pathways such as PI3K and PLCγ. This process activates NK cells to destroy target cells. Unlike some receptors, CD16a can activate NK cells independently of co-stimulatory signals. However, its surface expression diminishes after activation, presenting a limitation for sustained responses [7]. AFM13, an NKCE targeting CD16a, exemplifies a bispecific killer cell engager (BiKE) designed to overcome these challenges. Developed using the Redirected Optimized Cell Killing (ROCK) platform, AFM13 binds to a specific epitope on CD16a that is independent of Fc binding and remains unaffected by variations in patient CD16a allotypes. This ensures consistent efficacy even in the presence of serum IgG, which often interferes with Fc-mediated therapies [9,65].

#### 5.2.2. NKp46

NKp46 is a conserved glycoprotein regarded as a specific marker for human NK cells, being expressed on all mature NK cells. This receptor recognizes conserved tumor antigens and interacts with ITAM-containing molecules like CD3ζ and FcRγ to facilitate tumor cell recognition and NK cell activation. While NKp46 has low activation efficiency when acting alone, it is highly effective when co-activated with other receptors such as DNAM-1 and CD2. Advanced technologies, such as Antibody-based NK Cell Engager Therapeutics (ANKET), incorporate NKp46 as a target. These constructs combine Fab fragments targeting NKp46 and tumor antigens with Fc domains optimized for better binding affinity. These designs enhance tumor targeting and NK cell activation, broadening the scope of NKp46-based immunotherapies [7,66,67].

#### 5.2.3. NKp30

NKp30 is a type I transmembrane protein belonging to the immunoglobulin superfamily. Activation of NKp30 induces cytotoxic responses and the secretion of cytokines such as interferon-gamma (IFN-γ) and tumor necrosis factor-alpha (TNF-α). NKp30 binds to tumor-specific molecules, including B7-H6, which is rarely expressed on normal cells but highly expressed on tumor cell lines, conferring tumor specificity. Although NKp30 is a promising target, its low expression levels on NK cells may limit its activation potential. Therapeutic strategies aim to enhance NKp30 functionality to increase its contribution to antitumor responses [7,63].

#### 5.2.4. NKG2C

NKG2C is an activating receptor that forms a heterodimer with CD94 and binds to HLA-E on target cells. This interaction triggers downstream signaling through the adapter protein DAP12, initiating NK cell activation. Like CD16a, NKG2C can independently activate NK cells without requiring additional co-stimulatory signals. Its slower clearance rate compared to CD16a enables more sustained activation, making it an attractive target for NKCE therapies. Despite its advantages, the clinical utility of NKG2C is limited by its variable expression on NK cells. The frequency of NKG2C expression is influenced by a patient’s prior exposure to cytomegalovirus, which may restrict its therapeutic potential in certain populations [68,69].

#### 5.2.5. NKG2D

NKG2D, a member of the lectin family of activating receptors, triggers NK cell cytotoxicity and cytokine secretion. Unlike CD16a, NKG2D ligands are typically absent on healthy cells but appear under conditions of metabolic stress, irradiation, viral infection, or malignant transformation. This specificity allows NKG2D to rapidly detect and eliminate stressed or transformed cells. While its responses are faster than those mediated by CD16a, they are less potent. Targeting NKG2D ligands is critical for maintaining NK cell infiltration and functionality within the tumor microenvironment. Evidence shows that the absence of NKG2D ligands reduces NK cell motility, limiting their ability to target tumors effectively. This highlights the importance of therapies that sustain NKG2D-mediated responses [7,70].

These receptor-specific strategies highlight the versatility of NKCEs in utilizing diverse NK cell pathways to enhance tumor targeting and cytotoxic responses. By targeting receptors such as CD16a, NKp46, NKp30, NKG2C, and NKG2D, NKCEs provide innovative solutions to counter tumor immune evasion, offering improved therapeutic efficacy across a wide range of cancer types. As a result, CD16a, NKp46, NKp30, and NKG2C induce robust NK cell activation, promoting cytotoxicity as well as cytokine and chemokine secretion. In contrast, NKG2D primarily mediates cytotoxicity through DAP10-dependent PI3K signaling, leading to a relatively weaker activation of NK cells compared to the other receptors (Figure 5).

## 6. Phagocyte Engagers (PCEs)

PCEs are immunotherapeutic strategies that activate phagocytes such as macrophages and dendritic cells to eliminate tumor cells. These therapies use the innate immune system’s natural capacity for tumor clearance while counteracting immune evasion tactics used by cancer cells. A defining feature of PCEs is their ability to connect phagocytes directly to tumor cells, inducing antibody-dependent cellular phagocytosis or stimulating inflammatory responses [14,24]. One of the key mechanisms of PCEs involves targeting the CD47-SIRPα axis. Tumor cells frequently overexpress CD47, a “Don’t Eat Me” signal that binds to SIRPα on macrophages, inhibiting phagocytosis [14,71] (Figure 6). Blocking this interaction restores macrophage-mediated clearance of tumor cells, enhances antigen presentation, and shifts tumor-associated macrophages from an immunosuppressive M2 phenotype to a pro-inflammatory M1 phenotype, boosting antitumor immunity [72,73]. PCEs function as bispecific antibodies targeting specific tumor antigens or as antigen-independent agents that modulate the tumor microenvironment, making them effective in both solid and hematologic malignancies [14,74]. Bispecific Dendritic-T Cell Engager (BiCE) enhances anti-tumor immunity by targeting dendritic cells. By binding to CLEC9A, BiCE improves the migration and antigen-presenting capacity of conventional type 1 dendritic cells (cDC1). This promotes the formation of robust immune synapses with PD-1+ T cells, leading to increased IL-12 secretion and T cell activation, ultimately driving a potent and sustained anti-tumor immune response [75].

### 6.1. Macrophages

Macrophages, derived from monocytes originating in bone marrow, play a pivotal role in the innate immune system. They perform diverse functions including tumor cell phagocytosis, tissue remodeling, wound repair, modulation of inflammatory responses, and antigen presentation. Based on their activation state, macrophages are categorized as M1 or M2 phenotypes [76]. M1 macrophages are activated by signals such as interferon-gamma, toll-like receptor ligands, and pro-inflammatory cytokines. These macrophages secrete interleukin-1 beta and interleukin-12, which enhance inflammatory responses and promote tumor cell death. They also release chemokines like CXCL9 and CXCL10, which recruit cytotoxic T cells to the tumor site and amplify their activity [14,24]. In contrast, M2 macrophages are activated by anti-inflammatory signals such as interleukin-4, interleukin-10, and transforming growth factor-beta. M2 macrophages contribute to tissue repair and immune suppression, facilitating tumor growth and metastasis. Within the tumor microenvironment, macrophages are often co-opted as tumor-associated macrophages, which exhibit M2-like properties. Tumor-associated macrophages secrete factors that suppress T-cell activity, promote angiogenesis, and enhance tumor progression [14,24,77] (Figure 7).

### 6.2. Bispecific Macrophage Engagers (BiMEs) Target Receptors

BiMEs designed to enhance macrophage-mediated antitumor responses by targeting specific receptor-ligand interactions. BiMEs simultaneously bind tumor-associated antigens (TAAs) on cancer cells and activating receptors on macrophages, such as Fcγ receptors (FcγRI, FcγRIIa, and FcγRIIIa), to induce antibody-dependent cellular phagocytosis (ADCP) and cytokine secretion. Additionally, BiMEs target immune checkpoint pathways, such as the SIRPα/CD47 axis, where cancer cells suppress phagocytosis by delivering a “don’t eat me” signal. Blocking this interaction with HMBD004 antibody restores macrophage activity, promoting tumor clearance. Other receptors, such as FcαRI, which interacts with IgA domains, and co-stimulatory molecules like CD40, further enhance macrophage activation and stimulate adaptive immune responses by facilitating tumor antigen presentation to T cells. Preclinical studies show that BiMEs effectively link macrophages to tumor cells, overcoming immune suppression in the tumor microenvironment and driving robust antitumor activity. These advancements position BiMEs as promising next-generation cancer immunotherapies [78,79,80] (Figure 8).

### 6.3. Dendritic Cells

Dendritic cells, derived from bone marrow progenitors, are key antigen-presenting cells that bridge innate and adaptive immunity. Immature dendritic cells capture antigens and, upon maturation, migrate to lymph nodes where they activate cytotoxic T cells. Within the tumor microenvironment, dendritic cells often lose functionality due to immunosuppressive factors like TGF-β, VEGF, and interleukin-10, which inhibit antigen presentation and maturation [81,82,83]. Therapeutic strategies aim to restore dendritic cell function by blocking these suppressive pathways or promoting dendritic cell activation. Anti-VEGF therapies enhance dendritic cell differentiation and survival, while TGF-β inhibitors restore antigen presentation. The enzyme IDO, which depletes tryptophan to suppress T-cell activity, is also targeted to improve dendritic cell functionality [84,85]. Dendritic cell engagers are emerging therapies that link dendritic cells with T cells to enhance immune responses. By bridging cDC1 cells with PD-1+ T cells, dendritic cell engagers strengthen immune synapses, promoting robust antitumor immunity in preclinical models [75,86].

## 7. Market Trends

To date, nine FDA-approved and marketed products in the immune cell engager (ICE) category are exclusively T-cell engagers (TCEs). Natural killer cell engagers and phagocyte cell engagers entered research later than TCEs, and while several candidates are currently in clinical trials, none have yet reached commercialization. Teclistamab (Tecvayli), Epcoritamab (Epkinly), Talquetamab (Talvey), and Blinatumomab (Blincyto) are among the prominent TCEs, with robust market growth projections and increasing global reach. Teclistamab, approved in 2022, and Talquetamab, approved in 2023, are key players in multiple myeloma therapies, while Epcoritamab, also approved in 2023, enhances immuno-oncology research. Blinatumomab, the first FDA-approved TCE in 2014, continues to lead with notable sales growth, particularly in acute lymphoblastic leukemia [4]. Complementing these are newer therapies like Mosunetuzumab-axgb, Tebentafusp-tebn, Elranatamab-bcmm, Glofitamab-gxbm, and Tremelimumab, targeting various cancers with innovative mechanisms, underscoring the expanding role of immune cell engagers in oncology and their diverse therapeutic applications (Table 1).

### 7.1. Teclistamab (Tecvayli)

Teclistamab (Tecvayli), developed by Johnson & Johnson, is a T-cell engager designed for the treatment of multiple myeloma. Approved by the FDA on 25 October 2022, it has gained significant traction in cancer treatment and immunotherapy research. Its presence in the global market continues to expand, driven by increasing demand for advanced multiple myeloma therapies. Research institutions and pharmaceutical companies are actively incorporating Teclistamab into their studies, contributing to ongoing advancements in immuno-oncology. The global market for Teclistamab is expected to grow at a compound annual growth rate (CAGR) of 17.5% from 2024 to 2030.

### 7.2. Epcoritamab (Epkinly)

Epcoritamab (Epkinly), co-developed by AbbVie and Genmab, received FDA approval on 19 May 2023. As a novel TCE, it plays a key role in immuno-oncology research and cancer treatment, offering an innovative therapeutic approach for managing malignancies. With ongoing advancements in immunotherapy, Epcoritamab’s market presence continues to grow. The global market for Epcoritamab is projected to expand at a CAGR of 5.2% from 2024 to 2030.

### 7.3. Talquetamab (Talvey)

Talquetamab (Talvey), another TCE developed by Johnson & Johnson, was approved by the FDA on 9 August 2023. It has rapidly established itself as an important player in cancer immunotherapy, attracting attention for its contributions to improving cancer treatment outcomes. Support from biopharmaceutical companies has further strengthened its presence in the immunotherapy market. The global Talquetamab market is anticipated to grow at a CAGR of 5.9% from 2024 to 2030.

### 7.4. Blinatumomab (Blincyto)

Blinatumomab (Blincyto), developed by Amgen, was the first FDA-approved product in the immune cell engager category, receiving approval on 3 December 2014. It is highly effective in treating B-cell precursor acute lymphoblastic leukemia and continues to demonstrate strong market performance. In the third quarter of 2023, Blincyto sales increased by 55% year-over-year, reaching USD 220 million. This growth was driven by a 56% increase in volume and expanded use for patients with B-cell acute lymphoblastic leukemia. Sales growth was further supported by an 11% increase in the U.S., 12% in international markets, and a 27% rise in the Asia–Pacific region.

### 7.5. Mosunetuzumab-Axgb (Lunsumio)

Mosunetuzumab-axgb, developed by Genentech, is a bispecific antibody that targets CD20 and CD3. Approved by the FDA on 22 December 2022, it is designed for the treatment of relapsed or refractory B-cell non-Hodgkin lymphoma, marking a significant addition to the bispecific antibody landscape.

### 7.6. Tebentafusp-Tebn (Kimmtrak)

Tebentafusp-tebn, developed by Immunocore Limited, was granted FDA approval on 25 January 2022. It is the first T-cell receptor therapy approved for metastatic uveal melanoma, utilizing a novel mechanism that targets gp100, a melanoma-associated antigen.

### 7.7. Elranatamab-Bcmm (Elrexfio)

Elranatamab-bcmm, developed by Pfizer, received FDA approval on 14 August 2023. This bispecific antibody targets B-cell maturation antigen (BCMA) on multiple myeloma cells and CD3 on T cells, providing a promising treatment option for relapsed or refractory multiple myeloma.

### 7.8. Glofitamab-Gxbm (Columvi)

Glofitamab-gxbm, also developed by Genentech, was approved by the FDA on 15 June 2023. This bispecific antibody targets CD20 and CD3, offering an effective treatment for patients with relapsed or refractory diffuse large B-cell lymphoma (DLBCL).

### 7.9. Tremelimumab (Imjudo)

Tremelimumab, developed by AstraZeneca Pharmaceuticals, received FDA approval on 10 October 2022. This immune checkpoint inhibitor targets CTLA-4 and is used in combination with durvalumab for the treatment of unresectable hepatocellular carcinoma, representing a breakthrough in liver cancer immunotherapy.

## 8. Research Trends

To evaluate the current landscape of immune cell engager research, a total of 91 research articles published from 2019 to the present were identified through a PubMed search using “Immune cell engager” as the main keyword, with detailed searches including “NK cell engager”, “T cell engager”, and “Macrophage cell engager”. Among these studies, T-cell engagers (TCEs) were the most extensively studied, accounting for 51.6% of the publications. NK cell engagers (NKCEs) followed with 41.8% of the research focus, reflecting growing interest in leveraging innate immunity. Phagocyte cell engagers (PCEs) represented 6.6% of the studies, indicating their emerging status in the field. These proportions highlight the dominance of TCEs in the current research landscape while underscoring the increasing interest in NKCEs and the potential of PCEs as novel therapeutic approaches (Figure 9).

### 8.1. T-Cell Engagers (TCEs)

A total of 47 studies on T-cell engagers (TCEs) were analyzed, with bispecific T-cell engagers (BiTEs) representing 76.6% of these studies. Research on TCEs also includes investigations into CAR-T engagers (8.5%) and various novel types and mechanisms to enhance their efficacy and safety profiles. Among the targets explored, CD3 emerged as the most frequently studied, accounting for 12.8% of the research focus. The variety of targets explored in TCE studies highlights the wide therapeutic potential of this platform, with research encompassing a broad spectrum of malignancies. Solid tumors account for 10.6% of these studies, followed by 8.5% focused on AML [87,88]. This extensive body of research emphasizes the pivotal role of TCEs in driving advancements in immunotherapy across diverse cancer types (Figure 10).

### 8.2. NK Cell Engagers (NKCEs)

A total of 38 studies on NKCEs were analyzed. Among these, bispecific NK cell engagers (BiKEs) accounted for 55.3% of the research, followed by trispecific NK cell engagers (TriKEs) at 36.8%. While T-cell engagers are predominantly studied as bispecific antibodies (BsAbs), NKCE research demonstrates a higher proportion of studies on TriKEs, reflecting the interest in leveraging multispecificity to enhance NK cell activation. The scope of NKCE research spans a diverse range of targets and diseases. In contrast to TCEs, NKCE studies exhibit a relatively greater emphasis on hematologic malignancies (AML 10.5%, MM 10.5%) compared to solid tumors (2.6%). This focus highlights the distinct therapeutic potential of NKCEs in treating blood cancers (Figure 11).

### 8.3. Phagocyte Cell Engagers (PCEs)

Phagocyte cell engagers (PCEs) represent a relatively new area of research, with a total of six studies analyzed, reflecting their later development in the field. Among PCE studies, macrophages are the primary focus comprising 50.0% of the research. The targets of PCEs, such as CD47, SIRPα, PD-1, and PD-L1, are being developed in a balanced manner. PCE research places greater emphasis on solid tumors (50%) compared to hematologic malignancies (16.7%) (Figure 12).

## 9. Clinical Trends

Clinical trials were analyzed using clinicaltrials.gov and EU clinical trials, with “Immune cell engager” as the primary keyword and more specific searches including “T cell engager”, “TCE”, “NK cell engager”, “NKCE”, “PCE”, and “Macrophage cell engager”. We have gathered a summary of 22 clinical studies conducted between 2019 and 2024. Among the 22 clinical trials, T-cell engagers accounted for 54.5%, followed by NK cell engagers at 27.3%, and phagocyte cell engagers at 18.2% (Figure 13).

### 9.1. T-Cell Engagers (TCEs)

A total of 12 clinical trials involving T-cell engagers (TCEs) were anlayzed, with bispecific T-cell engagers (BiTEs) representing the predominant type. The majority of these trials (83.3%) focus on CD3 as a primary target on T cells, often in combination with multiple tumor-associated antigens (TAAs) to enhance specificity and efficacy. While most TCE trials are currently in Phase I (58.3%), a subset of studies has progressed to Phase II, either as parallel or concurrent trials (Figure 14).

### 9.2. Natural Killer Cell Engagers (NKCEs)

A total of six clinical trials involving NKCEs were analyzed. Bispecific NK cell engagers (BiKEs) dominate these trials, accounting for 66.7% of the studies. All NKCEs trials share a common approach of targeting CD16a on NK cells, typically in combination with tumor-associated antigens (TAAs) or other cellular receptors to enhance specificity and cytotoxic activity. At the clinical stage, none of the trials have progressed to standalone Phase II. Half of the trials (50.0%) are in Phase I, while the remaining 50.0% are in Phase I/IIa, reflecting the early-stage exploration of NKCE therapies in clinical development (Figure 15).

### 9.3. Phagocyte Cell Engagers (PCEs)

A total of four clinical trials involving phagocyte cell engagers (PCEs) have been analyzed. Unlike T-cell engagers (TCEs) and NK cell engagers (NKCEs), PCE trials primarily focus on immune cytokines (GM-CSF and Gal-3) rather than specific cell surface receptors, reflecting a unique therapeutic approach. Despite their later entry into clinical development compared to NKCEs, two PCE trials have progressed to Phase II, indicating their significant clinical potential and growing impact in the field (Figure 16).

## 10. Conclusions

Immune cell engagers (ICEs) represent a transformative advancement in cancer immunotherapy, bridging the innate and adaptive immune systems to target malignancies with precision and efficacy. These therapies, including T-cell engagers (TCEs), natural killer cell engagers (NKCEs), and phagocyte cell engagers (PCEs), have introduced novel mechanisms of action, such as MHC-independent tumor targeting, immune synapse formation, and modulation of the tumor microenvironment (TME) [4,8,14,86]. The ability of ICEs to engage specific immune cells, overcome immune evasion, and induce tumor-specific immune responses has positioned them as critical tools in modern oncology.

T-cell engagers, particularly bispecific T-cell engagers (BiTEs), dominate both research and clinical development, driven by the clinical success of blinatumomab and subsequent approvals like teclistamab and talquetamab. Advanced platforms, including half-life-extended BiTEs, checkpoint inhibitory T-cell engagers (CiTEs), and simultaneous multi-immune targeting engagers (SMiTEs), have addressed limitations such as short half-life, resistance, and limited efficacy in solid tumors [4,5,10,55]. These innovations have broadened the scope of TCE applications, enabling their use in diverse malignancies and making them a cornerstone of the ICE landscape.

NK cell engagers (NKCEs), though still in earlier stages of development, have shown promise in hematologic malignancies by leveraging the innate cytotoxic potential of NK cells. Bispecific NK cell engagers (BiKEs) and trispecific NK cell engagers (TriKEs) enhance NK cell-mediated cytotoxicity through activating receptors such as CD16a and NKG2D, with the addition of cytokines like IL-15 to boost efficacy. Despite challenges such as low NK cell infiltration in the TME, ongoing clinical trials reflect the growing interest in NKCEs as an alternative or complement to T-cell-based therapies, particularly in tumors with low immunogenicity [4,7,8].

Phagocyte cell engagers (PCEs) represent an emerging class of ICEs, targeting macrophages and dendritic cells to enhance tumor clearance. By modulating immune checkpoint pathways, such as the CD47-SIRPα axis, PCEs reprogram immunosuppressive tumor-associated macrophages (TAMs) into pro-inflammatory M1 phenotypes, promoting phagocytosis and antigen presentation [4,14,22,74,76,83]. The focus on solid tumors in PCE research highlights their distinct therapeutic potential, with ongoing trials exploring their efficacy in combination with other therapies to maximize antitumor responses.

Market trends underscore the growing impact of ICEs, with nine FDA-approved products as of 2023, primarily T-cell engagers. The success of blinatumomab and subsequent approvals of teclistamab, epcoritamab, and talquetamab exemplify the commercial and therapeutic viability of these therapies. While NKCEs and PCEs lag behind in clinical development, their unique mechanisms and potential to address unmet needs in oncology make them promising candidates for future breakthroughs.

Immune Checkpoint Engagers are promising cancer immunotherapies that target tumor-associated antigens while activating immune effector cells without relying on MHC recognition. However, their effectiveness is often limited by challenges such as cytokine release syndrome and tumor antigen escape. Cytokine release syndrome occurs when excessive cytokine production from overactivated immune cells triggers severe systemic inflammation. To regulate immune activation and minimize toxicity, various approaches are being explored, including gradual dose escalation, blockade of IL-6 signaling with treatments like tocilizumab, and modifications to Fc domains to reduce unintended immune responses. Tumor antigen escape, where cancer cells evade detection by altering or reducing the expression of target antigens, requires innovative solutions such as targeting multiple antigens simultaneously, combining ICEs with checkpoint inhibitors, and incorporating co-stimulatory molecules like 4-1BB and OX40 to strengthen immune responses and counteract resistance [89,90,91,92,93].

Advancements in antibody engineering, nanobody-based constructs, and the development of bispecific and trispecific antibodies continue to expand the potential of ICE therapies. Integrating ICEs with cytokine pathways has emerged as a promising strategy to improve immune cell persistence while reducing toxicity. Trispecific antibodies that include IL-15 have demonstrated the ability to enhance natural killer and T-cell proliferation, while tumor-localized IL-2 delivery helps activate immune cells directly at the tumor site with minimal systemic side effects. Additionally, emerging technologies such as oncolytic viruses, CAR-engineered immune cells, and targeted cytokine release systems are being developed to improve tumor infiltration and immune activation within the tumor microenvironment. These innovations play a crucial role in overcoming therapeutic resistance and optimizing the efficacy of ICEs for both hematologic and solid malignancies [11,12,23,48].

In conclusion, ICEs are reshaping the landscape of cancer immunotherapy, offering highly specific, scalable, and effective treatment options for a wide range of malignancies. The robust research efforts, expanding clinical trials, and growing market presence of ICEs reflect their potential to become a cornerstone of next-generation oncology therapeutics. As the field progresses, ICEs are poised to deliver on the promise of personalized, precision medicine, significantly improving outcomes for patients with both hematologic and solid tumors.

## Figures and Tables

**Figure 1 antibodies-14-00016-f001:**
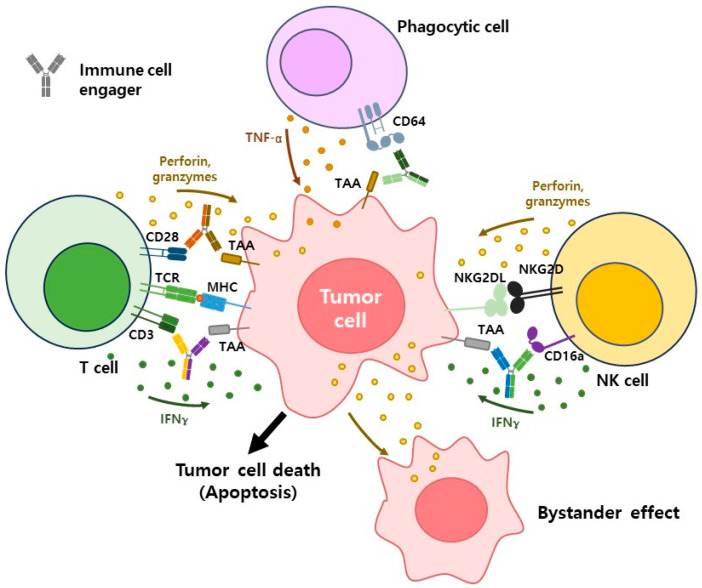
Mechanism of action of immune cell engagers (ICEs).

**Figure 2 antibodies-14-00016-f002:**
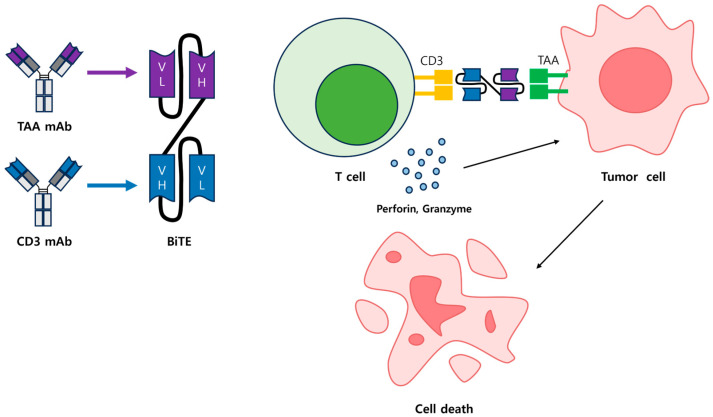
Structure of a BiTE and its tumor cell lysis mechanism. Illustrates the mechanism of action of bispecific T-cell engagers (BiTEs). A BiTE consists of two single-chain variable fragments (scFvs): one binds to CD3 on the surface of T cells, and the other binds to tumor-associated antigens (TAAs) on cancer cells. By simultaneously engaging CD3 and TAA, BiTEs bring T cells and cancer cells into close proximity, triggering a targeted cytotoxic response. During this process, T cells release perforin and granzymes, inducing apoptosis in cancer cells.

**Figure 3 antibodies-14-00016-f003:**
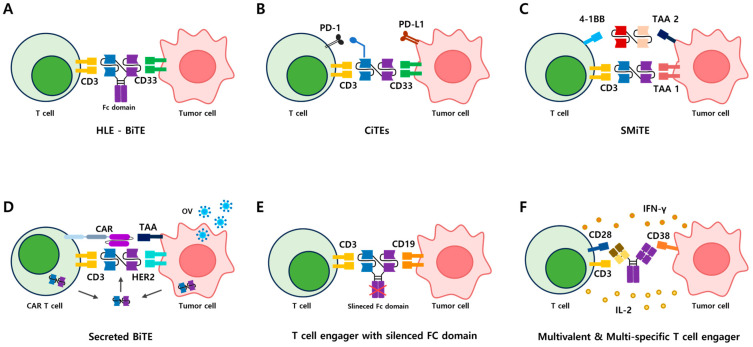
Modifications in BiTE structures and their functions. (**A**) HLE-BiTE: a half-life extended BiTE that includes an Fc domain to increase the drug’s serum half-life. It bridges CD3 on T cells and target antigens (TAA) on cancer cells, inducing cytotoxicity. (**B**) CiTEs: checkpoint inhibitory T-cell engagers that block the PD-1/PD-L1 pathway, combining immune checkpoint inhibition with T-cell activation. (**C**) SMiTE: stimulatory multi-specific T-cell engagers that target co-stimulatory receptors like CD28 on T cells to enhance activation and amplify immune responses. (**D**) CAR T cells and BiTE fusion: CAR T cells are engineered to secrete BiTEs, enabling direct attack on cancer cells while recruiting additional T cells to the tumor site, maximizing efficacy within the tumor microenvironment (TME). (**E**) T-cell engagers with silenced Fc domains: modified Fc domains are deactivated to prevent non-specific binding to Fcγ receptors, enhancing target specificity. (**F**) Multispecific and multivalent T-cell engagers: these engage multiple target antigens (CD38, TAA) and T-cell markers (CD3, CD28), inducing stronger and more comprehensive immune responses. This process triggers the secretion of IFN-γ and IL-2, boosting T cell activation and cytotoxicity.

**Figure 4 antibodies-14-00016-f004:**
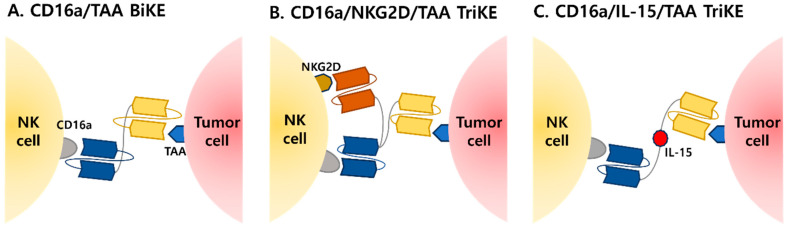
Representative structures of NK cell engagers (NKCEs). (**A**) BiKE targeting CD16a and TAA: the simplest NKCE structure, serving as the foundation for various derivatives. This design simultaneously recognizes the surface receptor CD16a on NK cells and tumor-associated antigens (TAA), narrowing the physical distance between cells and inducing antibody-dependent cellular cytotoxicity (ADCC). (**B**) TriKE Targeting CD16a and additional activating receptors: in addition to CD16a, this structure targets other activating receptors, such as NKG2D, providing co-stimulatory signals. This dual activation enhances NK cell activity and improves targeting precision. (**C**) TriKE with cytokine linker: An advanced BiKE structure targeting CD16a and TAA, with the addition of a cytokine (IL-15) as a linker. This design delivers cytokine stimulation alongside receptor targeting, inducing a robust cytotoxic response.

**Figure 5 antibodies-14-00016-f005:**
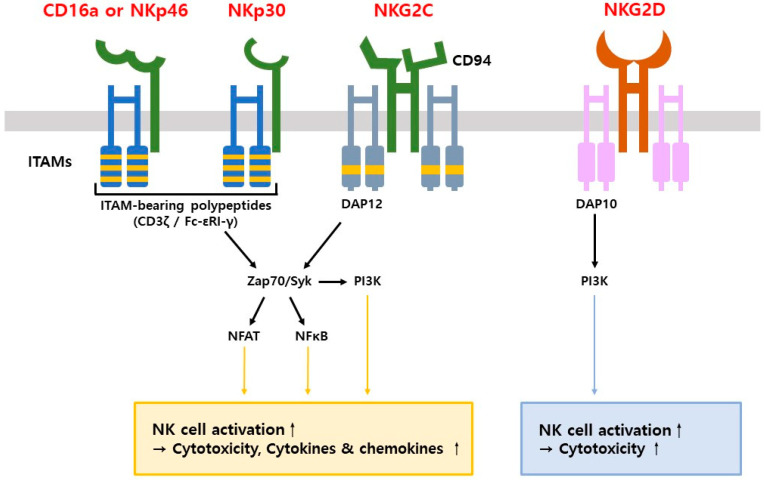
Schematic representation of receptors involved in NKCE mechanisms. NKCEs activate NK cells by targeting various receptors, including CD16a, NKp46, NKp30, NKG2C, and NKG2D. Structurally, CD16a, NKp46, and NKp30 are monomeric and classified as natural cytotoxicity receptors, whereas NKG2C and NKG2D are dimeric and categorized as C-type lectin receptors. NKG2C forms a heterodimeric structure with CD94, while NKG2D has a symmetric homodimeric configuration. Signal transduction is initiated through a signaling cascade starting with ITAM (immunoreceptor tyrosine-based activation motif) autophosphorylation. For CD16a, NKp46, and NKp30, ITAM function is mediated by CD3ζ/Fc-εRI-γ, whereas for NKG2C and NKG2D, it is mediated by DAP12 and DAP10, respectively.

**Figure 6 antibodies-14-00016-f006:**
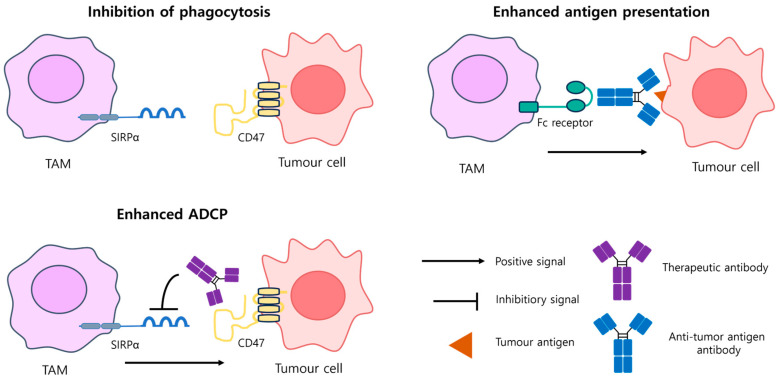
Overview of immune checkpoint receptors expressed by tumor-associated macrophages (TAMs) and ligands on tumor cells. TAMs play a key role in immune evasion and antitumor immunity through their interactions with immune checkpoint receptors and ligands. In the inhibition of phagocytosis, cancer cells evade immune detection by expressing CD47, which binds to the SIRPα receptor on macrophages, delivering a “don’t eat me” signal that suppresses phagocytosis. Therapeutic antibodies targeting this interaction restore macrophage phagocytic activity and promote cancer cell elimination. In the enhanced antigen presentation mechanism, TAMs recognize antibodies bound to tumor cells through Fc receptors, facilitating antibody-dependent cellular phagocytosis (ADCP). This process enables macrophages to process and present tumor antigens, activating T-cell-mediated immune responses and amplifying antitumor immunity.

**Figure 7 antibodies-14-00016-f007:**
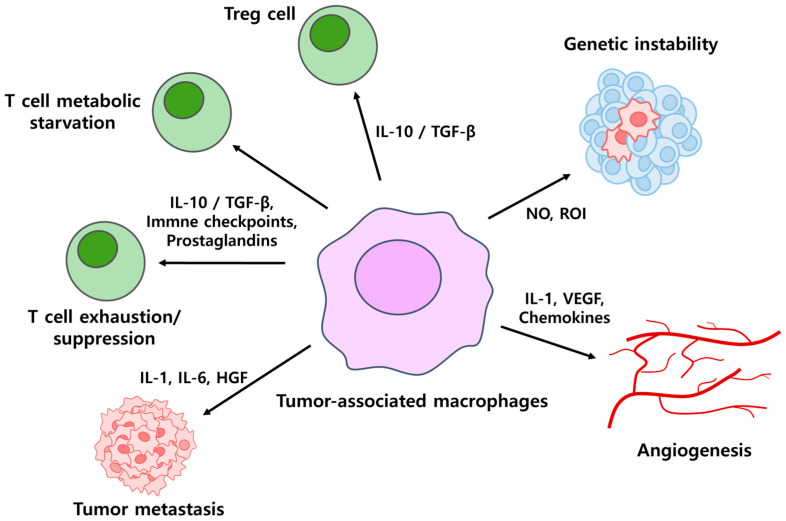
Functional diversity of tumor-associated macrophages (TAMs) and their mechanisms in tumor progression. TAMs contribute to tumor progression by secreting cytokines and chemokines that facilitate lymphocyte exclusion, promote angiogenesis, and induce tumor metastasis. TAMs release IL-10 and TGF-β to recruit regulatory T cells and inhibit normal T-cell immune responses by inducing lymphocyte exclusion through immune checkpoint blockade and prostaglandin secretion. TAMs produce nitric oxide (NO) and reactive oxygen intermediates (ROI), which contribute to DNA damage in tumor cells, leading to increased genetic instability and mutations. They also release reactive oxygen species (ROS), increasing mutation rates and aiding immune evasion. Additionally, TAM-derived IL-1, VEGF, and chemokines accelerate tumor development by promoting angiogenesis, while IL-1, IL-6, and HGF secretion facilitates tumor metastasis.

**Figure 8 antibodies-14-00016-f008:**
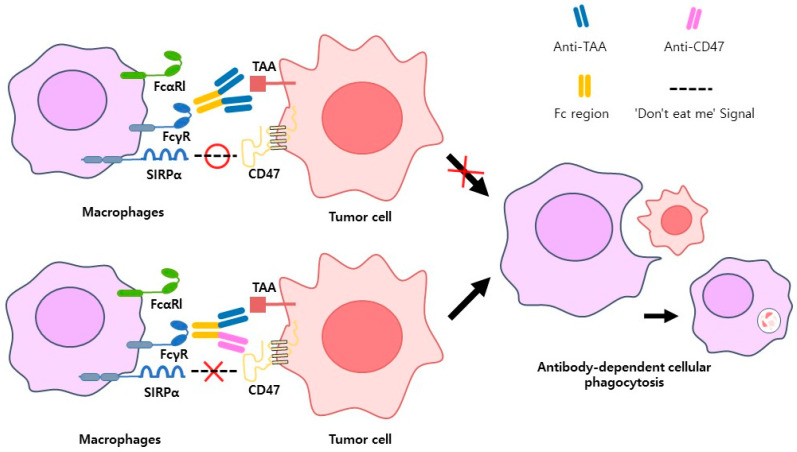
Antibody-dependent cellular phagocytosis (ADCP) mediated by macrophages. Antibodies targeting macrophages either block receptor-ligand pairs involved in immune checkpoints or recruit macrophages closer to tumor cells to promote ADCP. The FcγR receptor on macrophages binds to the Fc domain of antibodies, facilitating ADCP. Tumor cells evade immune responses by expressing CD47, which interacts with the SIRPα receptor on macrophages to deliver a “don’t eat me” signal. Antibodies binding to CD47 disrupt the SIRPα/CD47 axis, restoring phagocytic activity. The top left diagram illustrates how antibodies bind to tumor-associated antigens (TAA) and FcγR to promote ADCP. The bottom left diagram demonstrates how antibodies are designed to bind both FcγR and TAA while simultaneously blocking CD47 to enhance phagocytosis. Once tumor cells are engulfed by macrophages, phagosomes are formed, which fuse with lysosomes to create phagolysosomes, breaking down tumor cells.

**Figure 9 antibodies-14-00016-f009:**
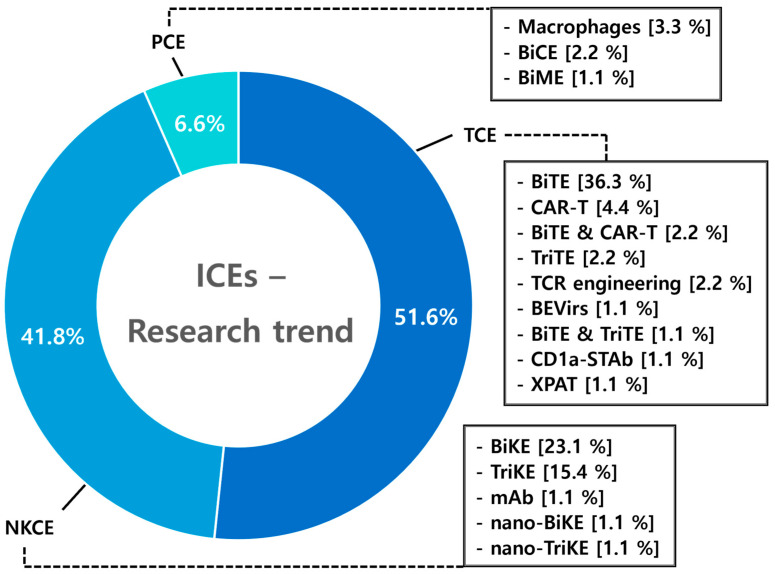
Emerging trend in immune cell engagers research. The distribution of research articles on immune cell engagers (ICEs) published from 2019 to 2025 is shown. Among 91 identified studies, T-cell engagers (TCEs) were the most frequently investigated, accounting for 51.6% of the publications. NK cell engagers (NKCEs) followed with 41.8%, indicating an increasing focus on innate immune mechanisms. Phagocyte cell engagers (PCEs) constituted 6.6% of the studies, reflecting their emerging status. These data highlight the dominant role of TCEs in the current research landscape, alongside the growing interest in NKCEs and the potential of PCEs as novel therapeutic modalities.

**Figure 10 antibodies-14-00016-f010:**
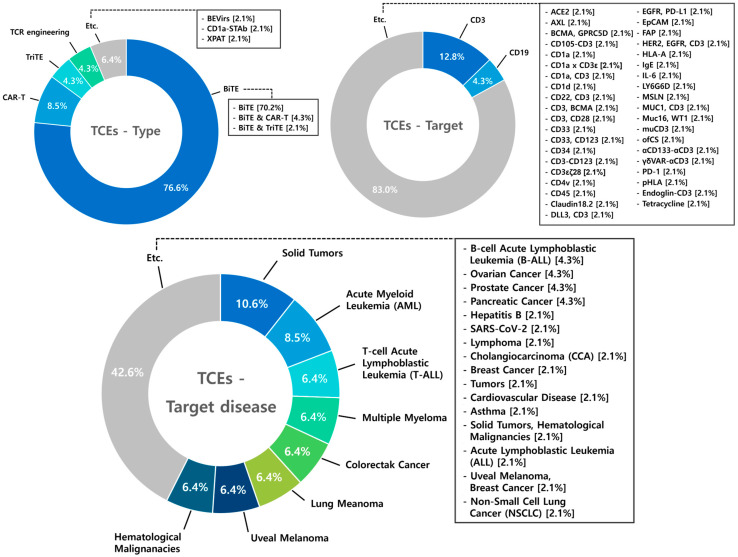
Diagram illustrating type, target, and target disease of T-cell engagers (TCEs). Among 47 studies on T-cell engagers (TCEs), bispecific T-cell engagers (BiTEs) dominate with 76.6%, followed by CAR-T engagers at 8.5%. Other novel types account for 15%, reflecting ongoing efforts to enhance the efficacy and safety profiles of TCEs. CD3 is the most frequently targeted molecule (12.8%), followed by other emerging targets such as CD19 (4.3%), indicating the diversity of strategies in TCE research. TCEs have been studied in solid tumors (10.6%), acute myeloid leukemia (AML, 8.5%), and other diseases, highlighting the broad therapeutic potential of TCE platforms.

**Figure 11 antibodies-14-00016-f011:**
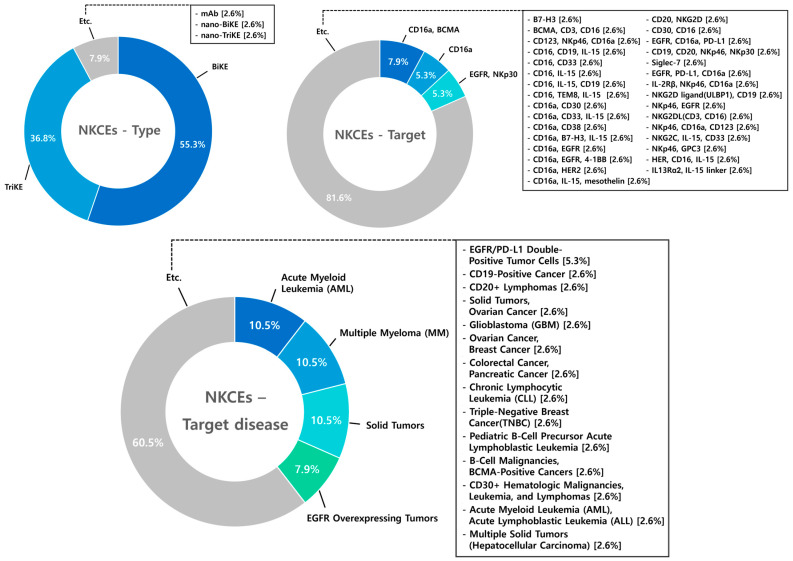
Diagram illustrating type, target, and target disease of NK Cell Engagers (NKCEs). BiKEs account for the majority of NKCE studies (55.3%), followed by TriKEs (36.8%) and other types (7.9%). Key targets include CD16a, BCMA (7.9%), CD16a (5.3%), EGFR, and NKp30 (5.3%), with other targets making up 81.6%, highlighting the diverse strategies explored to enhance NK cell activation. Target diseases for NKCEs include hematologic malignancies such as AML and MM (10.5% each), solid tumors (10.5%), EGFR overexpressing tumors (7.9%), and other diseases (60.5%), underscoring the broad therapeutic potential of NKCE platforms.

**Figure 12 antibodies-14-00016-f012:**
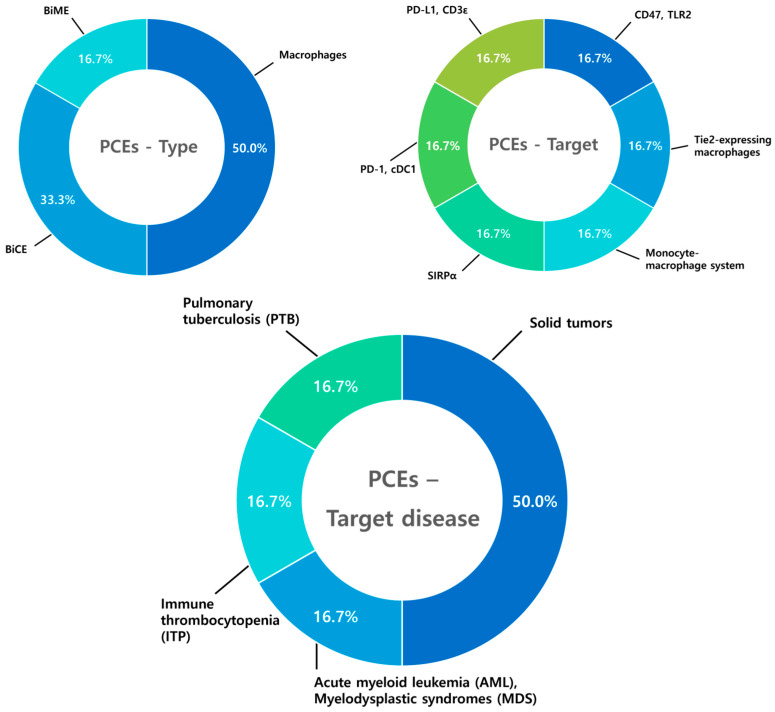
Diagram illustrating type, target, and target disease of phagocyte cell engagers PCEs. Among the six studies on phagocyte cell engagers (PCEs), macrophages account for 50.0% of the research, followed by BiCEs (33.3%) and BiMEs (16.7%). PCE targets are evenly distributed, with each target representing 16.7%, reflecting a balanced development of therapeutic strategies. PCEs place greater emphasis on solid tumors (50.0%) compared to hematologic malignancies and other diseases (16.7% each), indicating a focus on their potential application in oncology.

**Figure 13 antibodies-14-00016-f013:**
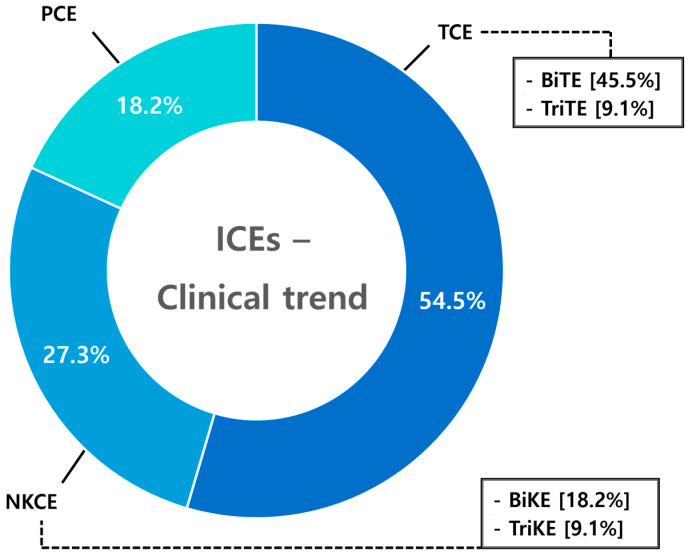
Clinical trends of ICEs. Clinical trial analysis identified 22 studies conducted between 2019 and 2024. Among these, T-cell engagers (TCEs) accounted for the majority (54.5%), with bispecific T-cell engagers (BiTEs) dominating, followed by trispecific T-cell engagers (TriTEs). NK cell engagers (NKCEs) represented 27.3% of trials, including bispecific NK cell engagers (BiKEs) and trispecific NK cell engagers (TriKEs). Phagocyte cell engagers (PCEs) accounted for 18.2%, highlighting their emerging presence in clinical exploration. This distribution illustrates the predominant clinical focus on TCEs, alongside the growing exploration of NKCEs and PCEs in advancing immune cell-based therapies.

**Figure 14 antibodies-14-00016-f014:**
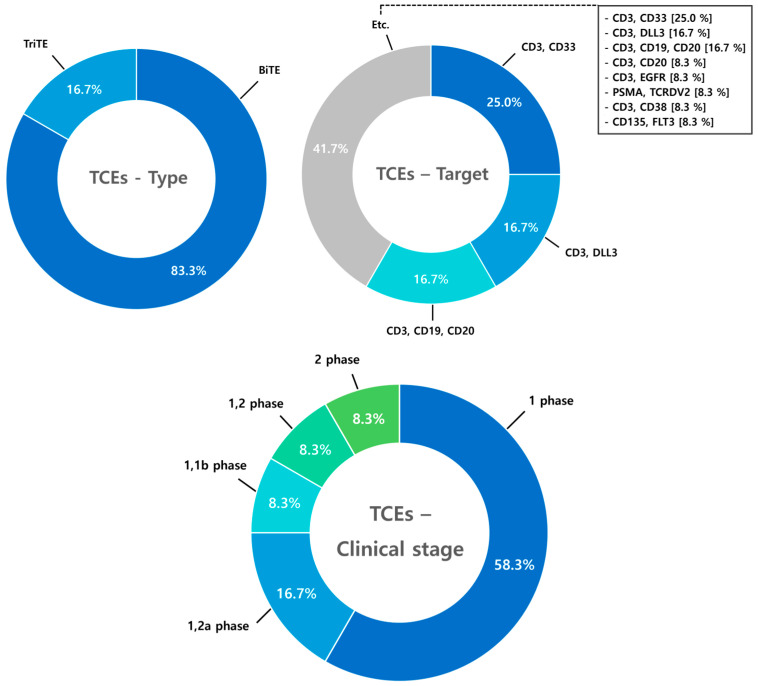
Clinical trends of TCEs. Analysis of 12 clinical trials on T-cell engagers (TCEs) reveals that bispecific T-cell engagers (BiTEs) dominate, representing 83.3% of the studied therapies. Most trials focus on CD3 as the primary target, often combined with tumor-associated antigens (TAAs) to enhance therapeutic efficacy. Phase I trials account for 58.3%, while 16.7% have advanced to Phase II. The remaining 25.0% comprise early-stage or exploratory studies.

**Figure 15 antibodies-14-00016-f015:**
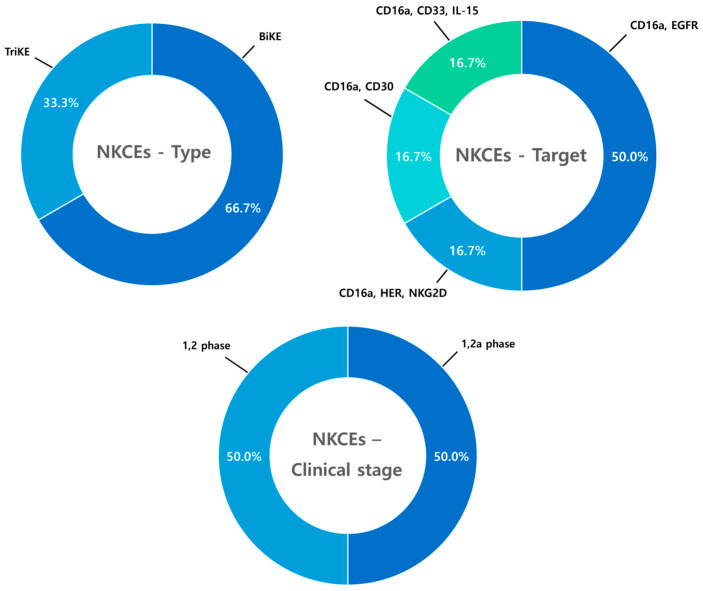
Clinical Trends of NKCEs. Analysis of six clinical trials involving NK cell engagers (NKCEs) reveals that bispecific NK cell engagers (BiKEs) dominate, accounting for 66.7% of the studies, while trispecific NK cell engagers (TriKEs) comprise 33.3%. All NKCE trials target CD16a, often in combination with tumor-associated antigens (TAAs) or additional cellular receptors, to enhance specificity and cytotoxic activity. At the clinical stage, 50.0% of the trials are in Phase I, while the remaining 50.0% are in combined Phase I/IIa studies. No standalone Phase II trials have been conducted, reflecting the early-stage exploration of NKCE therapies in clinical development.

**Figure 16 antibodies-14-00016-f016:**
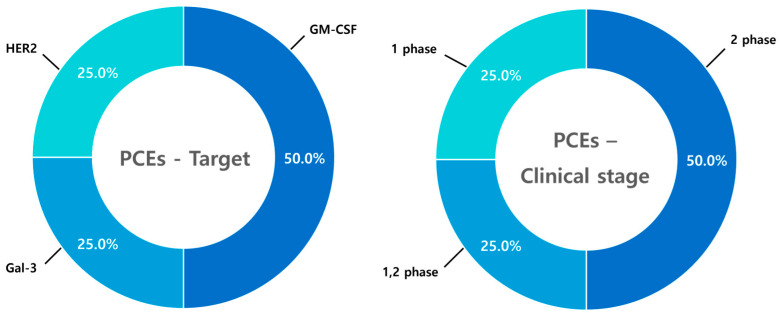
Clinical trends of PCEs. Among four clinical trials on phagocyte cell engagers (PCEs), 50.0% target immune-modulatory cytokines like GM-CSF, while Galectin-3 (Gal-3) and HER2 each account for 25.0%. Together, GM-CSF and Gal-3 represent 75.0% of the trials, reflecting a therapeutic strategy distinct from TCEs and NKCEs. At the clinical stage, 50.0% of PCE trials are in Phase I, while the remaining 50.0% have advanced to Phase II, indicating the increasing activity of PCE therapies in clinical development.

**Table 1 antibodies-14-00016-t001:** Overview of approved immune cell engagers (ICEs).

Drug Name	Company	ICEs Type	Target Antigen	Injection Type	Target Indication	Approval Date
Blinatumomab (Blincyto)	Amgen	TCE, biTE	CD3, CD19	Subcutaneous Injection	Acute Lymphoblastic Leukemia (ALL)	3 December 2014
Tebentafusp-tebn (Kimmtrak)	Immunocore Limited	TCE, biTE	CD3, gp100	Intravenous Injection	Uveal Melanoma	25 January 2022
Tremelimumab (Imjudo)	AstraZeneca Pharmaceuticals	Immune checkpoint inhibitor	CTLA-4	Intravenous Injection	Hepatocellular Carcinoma (HCC)	10 October 2022
Teclistamab (Tecvayli)	J&J	TCE, biTE	CD3, BCMA	Intravenous Injection	Relapsed or Refractory Multiple Myeloma	25 October 2022
Mosunetuzumab-axgb (Lunsumio)	Genentech	TCE, biTE	CD3, CD20	Intravenous Injection	Relapsed or Refractory Follicular Lymphoma	22 December 2022
Epcoritamab (Epkinly)	Abbvie and Genmab	TCE, biTE	CD3, CD20	Subcutaneous Injection	Relapsed or Refractory Large B-Cell Lymphoma	19 May 2023
Glofitamab-gxbm (Columvi)	Genentech	TCE, biTE	CD3, CD20	Intravenous Injection	Relapsed or Refractory Large B-Cell Lymphoma	15 June 2023
Talquetamab (Talvey)	J&J	TCE, biTE	CD3, GPRC5D	Subcutaneous Injection	Relapsed or Refractory Multiple Myeloma	9 August 2023
Elranatamab-bcmm (Elrexfio)	Pfizer	TCE, biTE	CD3, BCMA	Subcutaneous Injection	Relapsed or Refractory Multiple Myeloma	14 August 2023

## Data Availability

Not applicable.

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
