# Peer review of "Immune Cell Engagers: Advancing Precision Immunotherapy for Cancer Treatment"

_2073-4468, 2025, doi:10.3390/antib14010016_

Round 1

Reviewer 1 Report

Comments and Suggestions for Authors

The manuscript by In et al. reviews the mechanisms, antibody platforms, and clinical advancements of immune cell engagers (ICEs). The authors provide a comprehensive review of ICEs that engage immune effector cells such as T cells, NK cells, and macrophages. It highlights the success of T-cell engagers (TCEs), such as bispecific T-cell engagers (BiTEs), in hematologic malignancies. It also reviews recent innovation including trispecific antibodies and more advanced TCE platforms. In addition to T cells, NK cell engagers (NKCEs) and macrophage cell engagers (PCEs) are also discussed in detail. The manuscript outlines the challenges of ICEs, including their limited efficacy in solid tumors, resistance mechanisms, and safety concerns such as cytokine release syndrome.

This review provides a broad and accessible overview of ICEs and is an important resource for both researchers and clinicians in oncology. The manuscript can be further improved by expanding the discussion on challenges, specifically addressing the mechanisms underlying ICE-associated toxicity (e.g., cytokine release syndrome) and resistance (e.g., tumor antigen escape). It would also be intriguing if the authors explored potential strategies such as combining ICEs with cytokine signaling pathways.

Author Response

Response to the reviewers' comments

We thank the editor and reviewers for considering our study and we appreciate the time and care taken to provide us with valuable suggestions to improve the manuscript. We have addressed the specific concerns of the reviewers as indicated below and have revised our manuscript accordingly. We hope that our revised paper now meets the criteria for publication in Antibodies. 

Reviewer 1

The manuscript by In et al. reviews the mechanisms, antibody platforms, and clinical advancements of immune cell engagers (ICEs). The authors provide a comprehensive review of ICEs that engage immune effector cells such as T cells, NK cells, and macrophages. It highlights the success of T-cell engagers (TCEs), such as bispecific T-cell engagers (BiTEs), in hematologic malignancies. It also reviews recent innovation including trispecific antibodies and more advanced TCE platforms. In addition to T cells, NK cell engagers (NKCEs) and macrophage cell engagers (PCEs) are also discussed in detail. The manuscript outlines the challenges of ICEs, including their limited efficacy in solid tumors, resistance mechanisms, and safety concerns such as cytokine release syndrome.

  1. This review provides a broad and accessible overview of ICEs and is an important resource for both researchers and clinicians in oncology. The manuscript can be further improved by expanding the discussion on challenges, specifically addressing the mechanisms underlying ICE-associated toxicity (e.g., cytokine release syndrome) and resistance (e.g., tumor antigen escape). It would also be intriguing if the authors explored potential strategies such as combining ICEs with cytokine signaling pathways.

→ We appreciate the reviewer’s suggestions. We have revised the text on lines 809-833 and added the relevant references. The changes and additions have been marked in red.

Reviewer 2 Report

Comments and Suggestions for Authors

This manuscript is well-written and provides a thorough overview of ICEs, including their classifications, mechanisms of action, and emerging research trends. However, it can be further improved by addressing the following concerns:

  1. Figure 3 - Triple-Specific TCEs:
    The depiction of triple-specific TCEs targeting CD3, CD28, and CD38 in the treatment of myeloma is inaccurate. While activated T cells do express CD38, the design of these TCEs aims to bring T cells into proximity with myeloma cells, which express CD38. Please revise the figure to reflect this mechanism accurately. For reference, see the explanation provided here: Cancer.gov - Anti-CD38-CD28xCD3 Tri-Specific Monoclonal Antibody SAR442257.

  2. Figure 8 - Diagram Clarifications:

    • In the upper right panel, consider adding legends or annotations to clarify that the blue antibody recognizes the tumor-associated antigen (TAA).
    • In the lower left panel, distinguish the roles of each component for clarity: the red part of the scFv recognizes the TAA, the blue scFv targets CD47, and the Fc region (use a distinct color) binds to Fc receptors on macrophages. This will enhance reader understanding of the mechanism.

Author Response

Response to the reviewers' comments

We thank the editor and reviewers for considering our study and we appreciate the time and care taken to provide us with valuable suggestions to improve the manuscript. We have addressed the specific concerns of the reviewers as indicated below and have revised our manuscript accordingly. We hope that our revised paper now meets the criteria for publication in Antibodies.

Reviewer 2

This manuscript is well-written and provides a thorough overview of ICEs, including their classifications, mechanisms of action, and emerging research trends. However, it can be further improved by addressing the following concerns:

1. Figure 3 - Triple-Specific TCEs:

The depiction of triple-specific TCEs targeting CD3, CD28, and CD38 in the treatment of myeloma is inaccurate. While activated T cells do express CD38, the design of these TCEs aims to bring T cells into proximity with myeloma cells, which express CD38. Please revise the figure to reflect this mechanism accurately. For reference, see the explanation provided here: Cancer.gov - Anti-CD38-CD28xCD3 Tri-Specific Monoclonal Antibody SAR442257.
→ We appreciate the reviewer’s suggestions. we have modified Figure 3 as shown below.

(F) Multispecific and Multivalent T-cell Engagers: These engage multiple target antigens (CD38, TAA) and T-cell markers (CD3, CD28), inducing stronger and more comprehensive immune responses. This process triggers the secretion of IFN-γ and IL-2, boosting T cell activation and cytotoxicity.

2. Figure 8 - Diagram Clarifications:

  • In the upper right panel, consider adding legends or annotations to clarify that the blue antibody recognizes the tumor-associated antigen (TAA).
  • In the lower left panel, distinguish the roles of each component for clarity: the red part of the scFv recognizes the TAA, the blue scFv targets CD47, and the Fc region (use a distinct color) binds to Fc receptors on macrophages. This will enhance reader understanding of the mechanism.

    → In accordance with your suggestion, we have replaced figure 8 as shown below.

Reviewer 3 Report

Comments and Suggestions for Authors

The group of authors tried to review the mechanisms, platforms, research trends, and clinical progress of immune cell engagers. Due to the growing trials in the area and large market size, such a review seems to be a need in the area. The manuscript can be considered after revision

1-     The authors should define the search engines and keywords that were used to retrieve the articles

2-     In the platform section the authors must describe some details of the production

Author Response

Response to the reviewers' comments

We thank the editor and reviewers for considering our study and we appreciate the time and care taken to provide us with valuable suggestions to improve the manuscript. We have addressed the specific concerns of the reviewers as indicated below and have revised our manuscript accordingly. We hope that our revised paper now meets the criteria for publication in Antibodies.

Reviewer 3

The group of authors tried to review the mechanisms, platforms, research trends, and clinical progress of immune cell engagers. Due to the growing trials in the area and large market size, such a review seems to be a need in the area. The manuscript can be considered after revision

1. The authors should define the search engines and keywords that were used to retrieve the articles.
→ We appreciate the reviewer’s suggestions. We have modified the text on lines 706-708 as suggested. The changes and additions have been marked in red.

2. In the platform section the authors must describe some details of the production.
→ In accordance with your suggestion, we have modified the text on lines 570-633. The changes and additions have been marked in red.